# Mode of Antifungal Action of Daito-*Gettou* (*Alpinia zerumbet* var. *exelsa*) Essential Oil against *Aspergillus brasiliensis*

**DOI:** 10.3390/foods12061298

**Published:** 2023-03-18

**Authors:** Kiyo Okazaki, Hidenobu Sumitani, Katsutada Takahashi, Yuji Isegawa

**Affiliations:** 1Department of Health and Nutrition, Faculty of Human Life Science, Shikoku University, Furukawa, Ojin-cho, Tokushima 771-1192, Tokushima, Japan; 2Department of Food Sciences and Nutrition, School of Human Environmental Sciences, Mukogawa Women’s University, Nishinomiya 663-8558, Hyogo, Japan; 3Toyo Institute of Food Technology, Kawanishi 666-0026, Hyogo, Japan; 4Laboratory of Biophysical Chemistry, The Keihanna Academy of Science and Culture, Kyoto 619-0237, Kyoto, Japan; 5Department of Applied Biological Chemistry, Graduate School of Agriculture, Osaka Metropolitan University, Sakai 599-8531, Osaka, Japan

**Keywords:** *Alpinia zerumbet* var. *exelsa*, antifungal activity, Daito-*gettou*, essential oil, growth thermogram

## Abstract

Plant-derived essential oils (EOs) are used in medicines, disinfectants, and aromatherapy products. Information on the antifungal activity of EO of *Alpinia zerumbet* var. *exelsa* (known as Daito-*gettou*) found in Kitadaito Island, Okinawa, is limited. Therefore, we aimed to evaluate the antifungal activity of EOs obtained via steam distillation of leaves of Daito-*gettou*, which is a hybrid of *A. zerumbet* and *A. uraiensis*. Daito-*gettou* EO showed antifungal activity (minimum inhibitory concentration = 0.4%) against *Aspergillus brasiliensis* NBRC 9455, which was comparable to that of *A. zerumbet* found in the Okinawa main island. Gas chromatography/mass spectrometry revealed that the main components of Daito-*gettou* EOs are γ-terpinene, terpinen-4-ol, 1,8-cineole, 3-carene, and *p*-cymene. Terpinen-4-ol content (MIC = 0.075%) was 17.24%, suggesting that the antifungal activity of Daito-*gettou* EO was mainly attributable to this component. Daito-*gettou* EO and terpinen-4-ol inhibited mycelial growth. Moreover, calorimetric observations of fungal growth in the presence of Daito-*gettou* EO showed a characteristic pattern with no change in the initial growth rate and only a delay in growth. As this pattern is similar to that of amphotericin B, it implies that the action mode of Daito-*gettou* EO and terpinen-4-ol may be fungicidal. Further studies on the molecular mechanisms of action are needed for validation.

## 1. Introduction

Essential oils (EOs) derived from plants contain various components with antimicrobial activity [1,2,3,4,5,6,7], and they have long been used as disinfectants and medicinal agents. These EOs have also been used in food products to control spoilage by microorganisms and extend shelf life. In recent years, the emergence of bacteria resistant to chemical preservatives and the consumer preference for natural products have increased the demand for plant-derived EOs [8]. EOs derived from herbs or spices such as *Melaleuca alternifolia* (tea tree) [9,10], *Lavandula angustifolia* (lavender) [11], *Angelica major* [12], *Curcuma longa* L. [13], *Thymus capitatus* [14], *Thymus vulgaris* L. [15], and *Alpinia calcarata* Roscoe [12] have been reported to have antibacterial and antifungal activities. They are anticipated to inhibit the growth of food-spoilage microorganisms and maintain the quality and safety of food products.

In addition, the antimicrobial activity of EOs depends on the components of the EOs. These species contain numerous volatile compounds, such as terpenes and terpenoids, and aromatic and aliphatic components, which are naturally synthesized in various parts of plants as a part of their secondary metabolism [7]. The biological properties of EOs are determined by the concentration of their major components; therefore, analyzing the constituents of EOs is necessary to assess the biological activity. The components of EOs of *Syzygium aromaticum* L. Myrtaceae (clove) [16], *Citrus tangerina* (tangerine), *Carum carvi* (caraway) [17], and *Homalomena pineodora* [18] have been analyzed using gas chromatography/mass spectrometry (GC/MS).

*Alpinia zerumbet*, a perennial plant belonging to the genus *Alpinia* of the family Zingiberaceae, is commonly known as “shell ginger” in English and “*gettou*” in Japanese. It is distributed throughout tropical to subtropical Asia, and it is distributed from Okinawa Island to southern Kyushu in Japan. The seeds, leaves, and stems of this species have been used for various purposes [19]; for example, the seeds have been used as medicinal herbs since ancient times and are considered to have an intestinal-regulatory effect. Moreover, the stems are used in industries as a source of paper, and the leaves and rhizomes are frequently used as herbal teas and spices, respectively [19,20]. In Okinawa, it was customary to wrap foods with the leaves to carry around; the leaves are also used as a wrap for steaming buns and rice cakes. The leaves are still used to wrap the traditional rice cake *muchi*. Based on their experience, the people living in Okinawa, where the temperature and humidity are high and the foods spoil easily, have recognized that *gettou* has antiseptic and antibacterial properties. EOs or extracts of *gettou* leaves have antinociceptive [21], anti-inflammatory, and antipyretic activities [22]. Owing to their medicinal effects, EOs or extracts of *gettou* leaves have been used in skin care products and insect repellents. As EOs have a sweet scent, they are also used in aroma oils and fragrances.

There are multiple varieties of *gettou*, including Shima-*gettou* (*A. zerumbet*), which is widely distributed in Taiwan and the main island of Okinawa, Tairin-*gettou* (*A. uraiensis*) in northern Taiwan, and Daito-*gettou* (*A. zerumbet* var. *exelsa*) in Daito Islands, Okinawa Prefecture. Daito-*gettou* is considered to be a hybrid of *A. zerumbet* and *A. uraiensis* and has characteristics different from those of other *gettou*. For example, Daito-*gettou* grows to a height of over 4 m, which is taller than Shima-*gettou*, and does not bear fruit. The scent of Daito-*gettou* is similar to that of tea tree. Additionally, Daito-*gettou* when planted near sugarcane fields serves as a windbreaker and is used as a rope to bind harvested sugarcane.

In this study, the antifungal activity of EOs of the leaves of Daito-*gettou* distributed on Kitadaito Island was determined to investigate the relationship between the activity and chemical components of EOs. Tea tree EO was used for comparison in this study because it has a high antimicrobial activity [9], and its scent is similar to Daito-*gettou* EO; thus, it was postulated that the components might be similar between the two EOs. In addition, to further understand the mechanism underlying the antifungal action of Daito-*gettou*, we used a microbial calorimeter to measure the heat generation pattern (growth thermogram) associated with fungal growth in the presence of Daito-*gettou* EO or antifungal agents. We understood the effects of these agents on the growth of the fungi in detail. In this study, we aimed to determine the mechanism underlying the antifungal action of Daito-*gettou* EO by comparing its growth thermograms with those of common antifungal agents.

## 2. Materials and Methods

### 2.1. Essential Oils

Daito-*gettou* EO used in this study was donated by Kitadaito Island Development Organization, Okinawa Prefecture. Three Shima-*gettou* EOs were purchased from Nihon *Gettou* Co., Ltd. (Okinawa, Japan), Green Plan Shinjo Co., Ltd. (Okinawa, Japan), and Tree of Life Co., Ltd. (Tokyo, Japan) for comparison. These EOs are referred to as Shima-*gettou* 1, 2, and 3, respectively. Tea tree EO was purchased from Natural Organic Co., Ltd. (Tokyo, Japan).

### 2.2. Antifungal Agents and Chemicals

Miconazole, diflucan (fluconazole), and 5-fluorocytosine were purchased from Combi-Blocks Inc. (San Diego, CA, USA). Itraconazole and voriconazole were purchased from Tokyo Chemical Industry Co., Ltd. (Tokyo, Japan). 2-(4-Thiazolyl) benzimidazole (thiabendazole, TBZ) was obtained from San-ai Oil Co., Ltd. (Tokyo, Japan). Amphotericin B and terpinen-4-ol were purchased from FUJIFILM Wako Pure Chemical Corporation (Osaka, Japan), and α-pinene, camphene, 3-carene, limonene, γ-terpinene, 1,8-cineole, and *p*-cymene were purchased from Tokyo Chemical Industry Co., Ltd. (Tokyo, Japan).

### 2.3. Minimum Inhibitory Concentration

*Aspergillus brasiliensis* NBRC 9455 was purchased from the National Institute of Technology and Evaluation Biological Resource Center (NBRC, Tokyo, Japan) and used for antifungal tests. It was inoculated at three points on a Sabouraud dextrose agar plate containing 4.0% (*w*/*v*) glucose, 1.0% (*w*/*v*) peptone, and 1.5% (*w*/*v*) agar (Becton Dickinson, Sparks, MD, USA). After being incubated for 7 d at 25 °C, spore suspension was prepared by adding sterile saline (15 mL) containing 0.1% (*w*/*v*) polypeptone (Becton Dickinson) and 0.05% (*w*/*v*) Tween-80 (FUJIFILM Wako Pure Chemical Corporation, Osaka, Japan) to the plate and gently rubbing the surface of the fungal colonies with a sterile loop. The spore suspension was filtered through a coarse cloth to remove mycelia, and the spores were counted using a hemocytometer. The concentration of the spore suspension was adjusted to 2 × 10^5^ spores/mL using Sabouraud dextrose broth containing 4.0% (*w*/*v*) glucose and 1.0% (*w*/*v*) peptone for the broth dilution method.

The minimum inhibitory concentration (MIC) of EOs and antifungal agents for *A. brasiliensis* was determined using the broth dilution method. EOs and antifungal agents were dissolved in 20% ethyl alcohol solution. Ethyl alcohol was selected as the solvent in consideration of the future application of Daito-*gettou* to foods. Amphotericin B, which is insoluble in ethyl alcohol, was dissolved in DMSO (FUJIFILM Wako Pure Chemical Corporation). These solutions were serially diluted with Sabouraud dextrose broth, and 1 mL of the diluted solutions was pipetted into sterile test tubes containing 1 mL of the spore suspensions. The MIC was determined via visual inspection after incubating the mixtures at 25 °C for 3 d.

### 2.4. Gas Chromatography/Mass Spectrometry Analysis

The components of EOs were analyzed using gas chromatography/mass spectrometry (GC/MS). The instrument (JMS-T100GCV; JEOL Ltd., Tokyo, Japan) was equipped with an Agilent DB-5MS capillary column (30 m × 0.25 mm i.d., 0.25-μm film thickness, Agilent Technologies, Santa Clara, CA, USA). The oven was programmed as follows: initial temperature of 40 °C (held for 5 min) and heated at a rate of 10 °C /min to 320 °C (held for 3 min). The injector temperature was maintained at 250 °C. Helium was used as the carrier gas at a flow rate of 1.0 mL/min. The ionization voltage was 70 eV, and the mass range was 35–650 *m*/*z*. Analysis was performed using two methods. First, the mass spectrum of each EO component was determined by comparison with the mass spectrum from the NIST 11 spectrum library. The percentage composition of each component was calculated based on the respective peak areas. Second, some major components of the EO were quantitatively analyzed. The amount of each component was obtained from a calibration curve prepared using the corresponding standard product.

### 2.5. Effects on Spore Germination and Mycelial Growth

The inhibitory effects of Daito-*gettou* EO, Shima-*gettou* 1 EO, terpinen-4-ol, *p*-cymene, and 1,8-cineole on the mycelial growth of *A. brasiliensis* were determined using the modified “agar dilution method (2007)” for fungi [23]. Each antifungal agent was adjusted to concentrations at 25-, 50-, and 100-fold of the MIC with ethyl alcohol. The ethyl alcohol solutions (0.15 mL) and the dissolved Sabouraud dextrose agar medium (15 mL) were mixed in a Petri dish (φ90 mm) and solidified. The final concentrations of antifungal agents in agar media were adjusted to 1/4, 1/2, and 1/1 MIC, respectively. A sterile paper disk (Toyo Roshi Kaisha, Ltd. (Tokyo, Japan): φ13-mm thick) was placed on an agar plate, and 0.1 mL of the spore suspension (1 × 10^5^ spores/mL) was inoculated on the disk. After culturing at 25 °C for 3 d, the radius of the mycelia grown (excluding the radius of the paper disc) was measured, and the difference from the radius of the mycelia grown on an agar plate without antifungal agents (control) was used to estimate the inhibition of mycelial growth. The inhibitory rate was calculated as follows:

Mycelial growth inhibitory rate (%) = (1 − mycelial growth distance on agar medium containing antifungal agent/mycelial growth distance on control agar medium) × 100.

Furthermore, the inhibition of mycelial growth of *A. brasiliensis* by Daito-*gettou* EO and terpinen-4-ol was evaluated based on the weight of mycelia. Each solution of EO and terpinen-4-ol was serially diluted with Sabouraud dextrose broth, and 7.5 mL of the diluted solution was pipetted into a sterile Petri dish (φ90 mm) containing 7.5 mL of the spore suspension (2 × 10^5^ spores/mL with Sabouraud dextrose broth). The concentration of EO and terpinen-4-ol in the broth was adjusted to 0–0.4% or 0–0.075%, respectively. After incubating at 25 °C for 5 d, the mycelia in the cultures were filtered through a membrane filter (cellulose acetate, pore size: 0.8 μm) and washed with distilled water. Mycelia on the membrane filter were weighed after removing excess moisture with filter paper.

### 2.6. Calorimetric Measurements

A multiplex batch calorimeter, Leonis (ADVANCE RIKO, Inc., Yokohama, Japan), with 25 calorimetric units was used. The apparatus was developed by Japan Science and Technology, a government agency, and it is now sold as a commercial product termed “Non-destructive and Non-invasive Analytical Instrument,” which can quantitatively determine microbial growth activity [24,25,26]. As a sensor, semiconducting thermopile plates were employed and placed in an aluminum heat sink to detect the thermal change in the sample vessels (Petri dishes) set in each calorimetric unit. The calorimetric signals obtained were analyzed according to a method reported previously [25].

Petri dishes (φ60 mm) were placed on units as calorimetric vessels. When temperature changes occurred in the sample vessels, the sensor detected them, and the differentially generated voltage was proportional to the temperature changes. A mixed solution of 4.9 mL of the spore suspension (1 × 10^5^ spores/mL) prepared in Sabouraud dextrose broth and 0.1 mL of antifungal agent diluted stepwise with ethanol was poured into Petri dishes. The sample dishes were then placed in a calorimetric unit and maintained at 25 °C. Calorimetric output signals associated with fungal growth were monitored for incubation periods of 3–5 d.

### 2.7. Statistical Analysis

The results were statistically analyzed using a two-tailed unpaired Student’s *t*-test (Microsoft Excel 365; Microsoft Corporation, Redmond, WA, USA). Results with *p* < 0.05 were considered statistically significant.

## 3. Results

### 3.1. Antifungal Activity

The susceptibility of *A. brasiliensis* used in this study to various antifungal agents was measured before evaluating the antifungal activity of *gettou* EOs. The MICs of the antifungal agents are shown in Table 1. Three azol antifungal agents (MCZ, ITCZ, and VRCZ) and amphotericin B (AMPH-B), a polyene drug, showed high antifungal activities. Fluconazole (FLCZ) and flucytosine (5-FC), a pyrimidine-analog drug, presented lower activities. The antifungal agent TBZ (2-(4-thiazolyl) benzimidazole), which is widely used as a pesticide and food additive, had an MIC of 50 ppm (250 μmol/L). Table 1 also shows the MICs of EOs of various types of *gettou* and tea tree. Daito-*gettou* EO exhibited the same level of antifungal activity (MIC = 0.40%) as Shima-*gettou* 1–3 EOs, but it was less active than tea tree oil.

The GC/MS analysis of each EO component revealed that Daito-*gettou* contains γ-terpinene, terpinen-4-ol, 1,8-cineole, 3-carene, and *p*-cymene (Table 2, Figure 1). In contrast, the composition of Shima-*gettou* EO was quite different from that of Daito-*gettou*, and it was dependent on the product. Shima-*gettou* 1 and 2 EOs contained *p*-cymene, limonene, and α-pinene as the main components, with no traces of terpinen-4-ol. Tea tree EO contained γ-terpinene and terpinen-4-ol, but the peak area of terpinen-4-ol was considerably larger than that of the other EOs. These results are in agreement with those reported by Ninomiya et al. [10].

The antifungal activities of five compounds, which showed large peak areas in Daito-*gettou* EO, and three compounds (limonene, α-pinene, and camphene) detected in Shima-*gettou* 1 and 2 EOs were measured. As shown in Table 3, terpinen-4-ol showed the highest activity (MIC = 0.075%, 4.9 mmol/L) among the tested components. Therefore, terpinen-4-ol was considered as the major substance responsible for the antifungal activity of Daito-*gettou* EO. This result is consistent with those of Terzi et al. [27] and Roana et al. [28], who investigated the antifungal activity of tea tree oil. Terzi et al. described terpinen-4-ol as the most active component of tea tree oil against fungi. 1,8-Cineole also showed antifungal activity, with an MIC of 0.50% (32 mmol/L).

Table 4 shows the terpinene-4-ol and 1,8-cineol concentration/level of each EO. The concentration/level of *p*-cymene, which is a common component of all *gettou* EOs, is also shown. These components were quantified using the respective standards. The composition of Shima-*gettou* 1 and 2 EOs was significantly different from that of Daito-*gettou*.

### 3.2. Effects of Daito-Gettou EO on Mycelial Growth

To further understand the mechanism underlying the antifungal action of Daito-*gettou* EO, its effect on the mycelial growth of *A. brasiliensis* was measured. Shima-*gettou* 1 EO, terpinen-4-ol, *p*-cymene, and 1,8-cineole were used for comparison. The inhibitory rate increased with the increasing concentration (1/4 to 1/1 MIC) of all antifungal agents (Figure 2). When treated at 1/1 MIC, the inhibitory rate was 63.8% for Daito-*gettou* EO, 59.7% for Shima-*gettou* 1 EO, and 100% for terpinen-4-ol. Mycelial growth was observed despite treatment at the MIC, which might mainly be attributed to the obtained MIC value (Table 3), which was determined using the broth dilution method in this study. As mycelial growth inhibition was measured using the agar medium dilution method in which the antifungal agent was added to the agar medium and the MICs measured using the broth medium dilution method and agar medium dilution method may differ, the experimental results were believed to be inconsistent. In contrast, treatment at 1/4 and 1/2 MIC of *p*-cymene and 1/4 MIC of 1,8-cineole resulted in negative inhibition rates. In the presence of these compounds at MIC, mycelial growth was inhibited, whereas at low concentrations, growth was promoted.

The effects of Daito-*gettou* EO and terpinen-4-ol on mycelial weight were measured in Sabouraud dextrose broth medium. The addition of 0.2% EO to the medium increased the dry mycelial mass of the tested fungi, but the mass decreased as the EO concentration increased (Figure 3a). Furthermore, when 0.4% EO (equal to the MIC) was added, some amount of mycelium growth was observed. Additionally, only a few EOs have been shown to inhibit fungal spore germination and mycelial growth [15]. Pereira et al. [29] have also reported that the EO of *Cymbopogon winterianus* inhibits the mycelial growth of *Trichophyton rubrum*. It has also been shown that the vaporous phase of the EO of *Thymus vulgaris* L. (thyme) strongly suppresses the sporulation of fungi in glass chambers [30]. In this study, terpinen-4-ol, the main component of Daito-*gettou* EO, was measured in the same way (Figure 3b). Terpinen-4-ol at 0.075% (equal to the MIC) completely inhibited mycelial growth.

### 3.3. Growth Thermogram of A. brasiliensis in the Presence of Daito-Gettou EO

Figure 4a–d show the growth thermograms for cultures of *A. brasiliensis* in Sabouraud dextrose broth containing various concentrations of TBZ, AMPH-B, Daito-*gettou* EO, and terpinen-4-ol, respectively. These figures are termed “*g*(*t*) curves.” The vertical axis of each figure represents the thermoelectromotive force indicated by the heat detector (µV), and the horizontal axis represents the incubation time (min). The heat generated in the broth containing each sample is considered to be the metabolic heat associated with the process of fungal spore germination and the subsequent mycelial growth [31]. When spores of *A. brasiliensis* and TBZ at concentrations below the MIC were mixed in Sabouraud dextrose broth and cultured, the pattern of the growth thermogram changed as the TBZ concentration increased, and the slope of the *g*(*t*) curve decreased. Moreover, it was observed that the apparent peak time shifted to the longer side of the culture. Additionally, a delay in growth was observed. Thus, changes in the initial rate and delay in growth time occurred, and TBZ was found to suppress fungal growth in a concentration-dependent manner. In contrast, the growth of *A. brasiliensis* in the presence of AMPH-B was delayed, but no apparent change was observed in the initial growth rate. As shown in Figure 4c,d, the growth thermogram of Daito-*gettou* EO was similar to that of terpinen-4-ol. Our results revealed that there were time delays in the fungal growth as the concentrations of Daito-*gettou* EO or terpinen-4-ol increased; however, their initial growth rates did not change. The thermogram pattern of both Daito-*gettou* EO and terpinen-4-ol was closer to that of AMPH-B than to the pattern of TBZ.

An increase in microbial growth activity can be observed when growth thermograms are transformed into calorimetrically defined growth curves [24,32,33,34]. The growth curves described as “*f*(*t*) curves” were obtained by computation based on Equation (1):*f*(*t*) = *g*(*t*) + *K* ∫*g*(*t*) d*t*(1)
where *t* is the growth time and *K* is the heat conduction constant (Newton’s cooling constant) of each calorific value measuring unit including the Petri dish containing the culture solution [24,25,26,35]. The obtained *f*(*t*) curves are shown in Figure 5, which correspond to the growth curves of microorganisms in the presence of different agents. As shown in Figure 5a, we observed that the slope of the curve decreased and the growth rate slowed as the TBZ concentration increased. In contrast, the slope of the curve in Figure 5b did not change even when the concentration of AMPH-B increased, and only a delay in fungal growth was observed. Moreover, when the concentrations of Daito-*gettou* EO were 0.025% and 0.05%, the slope of the curves was almost similar to that of the curve without EO. However, at concentrations above 0.1%, the slope of the curves was greater than that of the curve without EO. The *f*(*t*) curve of Daito-*gettou* EO exhibited a unique shape, but it more closely resembled the characteristics of the AMPH-B curve than those of the TBZ curve. In addition, as shown in Figure 5d, the slope for terpinen-4-ol increased with the drug concentration, indicating a time lag. The *f*(*t*) curve of Daito-*gettou* EO showed the characteristics of an AMPH-B curve and was notably similar to that of the terpinen-4-ol.

## 4. Discussion

*Aspergillus brasiliensis*, a common fungus in the soil, often contaminates food. It is a commonly used fungal strain for the preservative efficacy test of ISO11930 (International Organization for Standardization, 2012), the Japanese Pharmacopoeia (2021), and the Standards for Food and Food Additives (Ministry of Health and Welfare Notification No. 370, 1959, Japan). In this study, we evaluated the activity of Daito-*gettou* EO against *A. brasiliensis*. The susceptibility of *A. brasiliensis* used in this study was first confirmed before evaluating the antifungal activity of *gettou*. The trends of the MICs of antifungal agents were consistent with those reported previously [36,37].

The antifungal activity of Daito-*gettou* EO against *A. brasiliensis* was found to be similar to that of Shima-*gettou* EOs (MIC = 0.40%), but it was lower than the activity of tea tree EO (Table 1). In addition, the GC/MS analysis revealed that the composition of Daito-*gettou* EO was different from that of Shima-*gettou* EOs (Table 2). Similar to tea tree oil, Daito-*gettou* EO contained a large amount of terpinen-4-ol (Table 3). Generally, external effects cause changes in the ratios of the constituents of EOs [38]. The composition of *gettou* EOs is known to fluctuate under the influence of climate parameters (temperature and precipitation) [39]. Moreover, the activity thereof differs depending on the production area or the variety of *gettou*. According to Ramos et al. [40], the major constituents of *A. zerumbet* var. *variegata* EO are 1,8-cineole (39%), β-pinene (11%), and β-caryophyllene (10%). As Daito-*gettou* EO used in this study contained a lesser amount of *p*-cymene and was abundant in terpinen-4-ol, this result supports the concept that the variety of *gettou* inhabiting the Daito Islands is different from that present in Okinawa main island.

Terpinen-4-ol, which was abundant in Daito-*gettou* EO, exhibited antifungal effects (MIC = 0.075%) against *A. brasiliensis* (Table 3). As the MIC of Daito-*gettou* EO, which contains approximately 17% of this component, was 0.40%, the activity of Daito-*gettou* EO could mainly be attributed to terpinen-4-ol. Moreover, Maior et al. [41] and Ninomiya et al. [42] have reported that terpinene-4-ol possess antifungal activity. These findings strengthen our claim of terpinene-4-ol being the main component responsible for the antifungal action of Daito-*gettou* EO. Additionally, regarding the activity of *p*-cymene, which is another component, Aznar et al. [43] reported that it completely inhibits the growth of *Candida lusitaniae* for at least 21 d at concentrations above 1 mmol/L at 25 °C. However, MIC of *p*-cymene against *Rhizopus oryzae* is >1024 μg/mL, indicating a poor antifungal effect [15]. Daferera et al. [44] postulated that the antifungal activity of EOs is primarily due to their major components; however, other phenomena, such as synergy and antagonism with minor components, are also possible. These findings suggest that the antifungal activity of Daito-*gettou* EO is most likely attributable to terpinen-4-ol and the combined effect of other components, such as 1,8-cineole and *p*-cymene. The activity of Shima-*gettou* 1 and 2 EOs, which are terpinen-4-ol-free, was considered to be mediated by α-pinene and limonene. However, we believe that these components alone cannot explain the antifungal activity of Shima-*gettou* EOs. It is likely that unknown antifungal components or their synergy with other components are involved in the antifungal activity of Shima-*gettou* EOs.

Moreover, the mechanism underlying the antifungal action of Daito-*gettou* EO was investigated by measuring its effects on the mycelial growth of *A. brasiliensis*. Daito-*gettou* EO showed an inhibitory effect on mycelial growth at different MICs (Figure 2 and Figure 3). Terpinen-4-ol also exerted a significant effect on mycelial growth. This result suggests that Daito-*gettou* EO inhibits mycelial growth, which could mainly be attributed to terpinen-4-ol.

Furthermore, we aimed to elucidate the mechanism underlying the antifungal action in detail. To achieve this, we used a microbial calorimeter to obtain growth thermograms of fungi growing in the presence of various antifungal agents. Growth thermograms showed different heat generation patterns depending on the action of antifungal agents. One of the authors of this study, Takahashi [45], had reported the differences in growth thermograms between bactericidal and bacteriostatic agents. According to this report, imidazolidinyl urea, which shows bactericidal action, caused a delay in the rise of the growth thermogram, but it did not alter the initial growth rate. Propylparaben, which exhibits bacteriostatic action, caused a change in only the initial growth rate. The bactericidal or bacteriostatic action of antimicrobial agents are not classified properly, and most drugs are considered to be bacteriostatic up to a certain concentration and bactericidal above that concentration. Therefore, several drugs are considered to exhibit both bacteriostatic and bactericidal activities. However, we believe that the information obtained from growth thermograms will provide key clues for investigating the mechanisms of action of antifungal drugs.

A mechanism of action of TBZ has been reported by Allen and Gottlieb [46]. Per their findings, TBZ inhibits the terminal electron transport system of mitochondria and exhibits highly selective toxicity against fungi. Additionally, Kano et al. [31], by measuring the growth thermograms of TBZ, showed that TBZ inhibits the mycelial growth of *Aspergillus niger* in a concentration-dependent manner. In contrast, AMPH-B, a polyene macrolide antibiotic that selectively binds to ergosterol in cell membranes, is known to disrupt cell membranes, leak intracellular substances, and kill cells [47]. Based on these reports, the theory to quantitatively characterize the antimicrobial action of drugs [45], and the pattern of the *f*(*t*) curves obtained in this study, it can be hypothesized that TBZ exerts a fungistatic effect at low concentrations and exhibits fungicidal activity at high concentrations, and that AMPH-B is a fungicidal agent. Furthermore, we analyzed the *f*(*t*) curve of Daito-*gettou* EO and found that there was little change in the slope of the curve between concentrations of 0% and 0.05%, and the slope increased with the increase in concentration (>0.1%). The antifungal action of Daito-*gettou* EO did not change the initial growth rate but caused only a delay. Therefore, these results suggest that the antifungal action of Daito-*gettou* EO is fungicidal. Additionally, the *f*(*t*) curve of Daito-*gettou* EO was notably similar to that of terpinen-4-ol as shown in Figure 5, thereby implying that the fungicidal action of Daito-*gettou* EO is mainly due to the action of terpinen-4-ol.

In this study, the antifungal activity of Daito-*gettou* EO was compared by focusing on terpinen-4-ol, which is one of the main components. Terpinen-4-ol is a hydrophobic terpene that can strongly interact with microbial membrane lipids and affect membrane permeability. Polec et al. [48] reported that the antifungal activity of terpinen-4-ol is directly related to its incorporation into cellular membranes and is affected by the lipid composition of various pathogenic membranes. Furthermore, Li et al. [49] mainly attributed the antimicrobial activity of tea tree oil to the presence of terpinen-4-ol, and tea tree oil penetrated the cell wall and cytoplasmic membrane of the tested bacterial and fungal strains. In addition, their findings suggest that tea tree oil also penetrates fungal organelle membranes and exerts its antimicrobial effects by compromising the cell membrane, resulting in cytoplasm loss and organelle damage, which ultimate lead to cell death. These reports support the results of this study regarding the antifungal activities of Daito-*gettou* EO and terpinen-4-ol. It is presumed that Daito-*gettou* EO and terpinen-4-ol interact with the cell membrane of *A. brasiliensis*, affect the fluidity and permeability of the cell membrane, and substantially inhibit the growth of mycelia. Furthermore, Daito-*gettou* EO and terpinen-4-ol might disrupt cell membranes and cause intracellular substance leakage, and thereby exert a fungicidal effect against *A. brasiliensis*. However, it is apparent that the content of terpinen-4-ol in Daito-*gettou* EO is lower than that in tea tree EO, as determined using GC/MS analysis, and that Daito-*gettou* EO contains many other trace components in addition to terpinen-4-ol. Therefore, further investigations are necessary to clarify the detailed mechanism of antifungal action of Daito-*gettou* EO.

Daito-*gettou* EO has been closely associated with the life of people inhabiting Okinawa and the Daito Islands by supporting a safe food environment. It is used extensively and is considered to be a proxy for guaranteed food safety. Daito-*gettou* EO is considered particularly useful as a natural preservative for food to maintain hygiene. However, it is assumed that the EO composition of *gettou* fluctuates with the climate factors (such as temperature and rainfall) [39], which can lead to a corresponding fluctuation in its antifungal activity. In addition, Daito-*gettou* EO may contain unknown antifungal components, similar to those of Shima-*gettou* EOs, and therefore, further studies (for example, studies on synergistic effects between EO components and the influence of EO extraction methods) are required to elucidate the mechanisms underlying this activity.

## 5. Conclusions

In this study, the mechanism underlying the antifungal action of Daito-*gettou* EO was investigated, and the following major results were obtained. (1) Daito-*gettou* EO showed an antifungal effect against *A. brasiliensis* (MIC = 0.4%). (2) The main chemical components of EO were identified as γ-terpinene, terpinen-4-ol, 1,8-cineole, 3-carene, and *p*-cymene, which differed from the three kinds of Shima-*gettou* EOs used in this study. (3) Terpinen-4-ol, which is present in Daito-*gettou* EO at 17.24%, showed a higher antifungal activity than the other components (MIC = 0.075%), and the activity of Daito-*gettou* EO against *A. brasiliensis* could be attributed to this component. (4) Daito-*gettou* EO inhibited mycelial growth. (5) The pattern of growth thermograms, which were calorimetric observations of fungal growth in the presence of Daito-*gettou* EO, was similar to that of the fungicide amphotericin B. These findings imply that the mode of action of Daito-*gettou* EO is fungicidal; however, to confirm this, further studies on the molecular mechanisms of action are needed.

## Figures and Tables

**Figure 1 foods-12-01298-f001:**
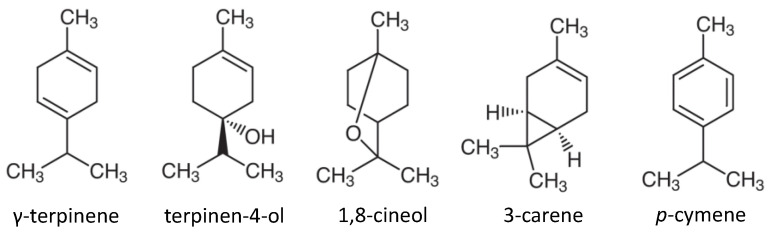
Chemical structures of the main compounds of Daito-*gettou* EO.

**Figure 2 foods-12-01298-f002:**
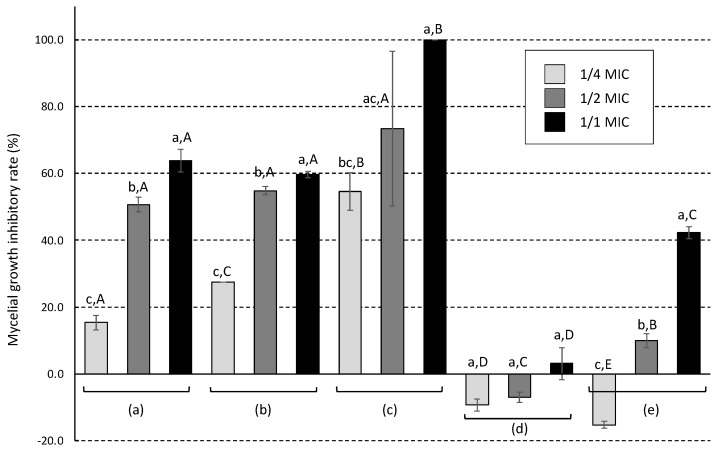
Effect of Daito-*gettou* EO (**a**) and Shima-*gettou* 1 EO (**b**), terpinen-4-ol (**c**), *p*-cymene (**d**) and 1,8-cineole (**e**) on the mycelial growth of *A. brasiliensis* NBRC 9544. The fungal spores were cultured on the agar plate containing each antifungal agent with the concentration of 1/1 or 1/2 or 1/4 MIC values at 25 °C for 3 d, and then, the inhibitory rate was calculated as follows: Mycelial growth inhibitory rate (%) = {1 − (mycelial growth distance on agar medium containing antifungal agent)/(mycelial growth distance on control agar medium) × 100}. Data with different letters (a–c) between the same antifungal agents are significantly different (*p* < 0.05). Data with different letters (A–E) between antifungal agents are significantly different (*p* < 0.05).

**Figure 3 foods-12-01298-f003:**
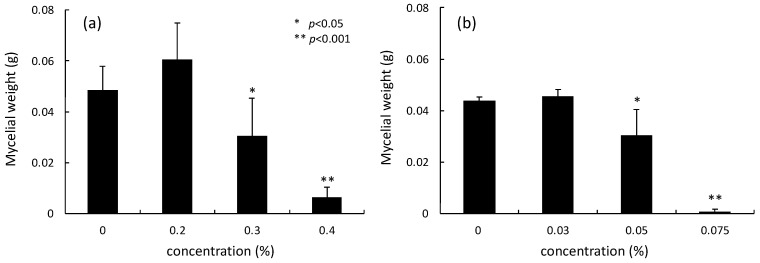
Effect of different concentrations of Daito-*gettou* EO (**a**) and terpinen-4-ol (**b**) on dry mycelial weight.

**Figure 4 foods-12-01298-f004:**
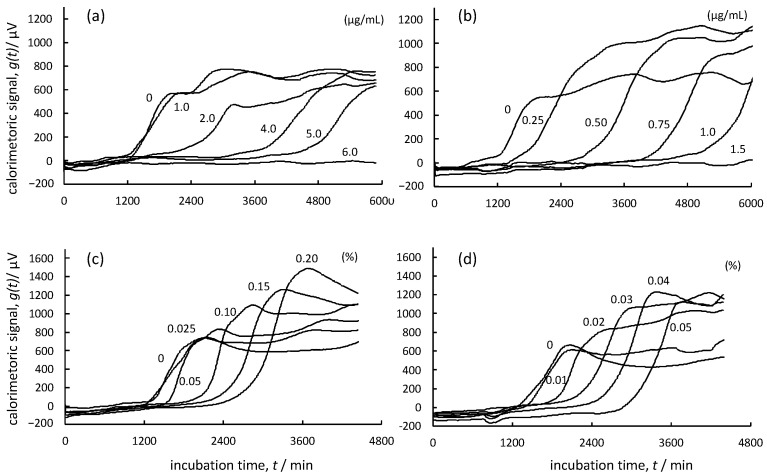
Growth thermograms of *A. brasiliensis* NBRC 9544 grown at 25 °C in Sabouraud dextrose broth containing TBZ (**a**), amphotericin B (**b**), Daito-*gettou* EO (**c**), and terpinen-4-ol (**d**) at various concentrations. The concentration “0” in each figure shows the growth thermogram of *A. brasiliensis* in the broth without antifungal agents.

**Figure 5 foods-12-01298-f005:**
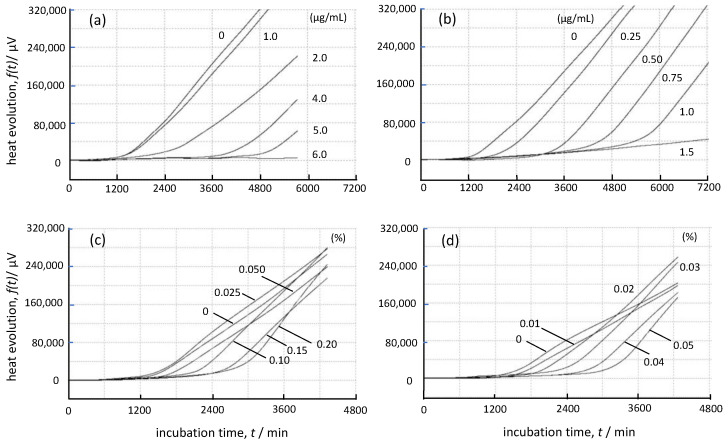
*f*(*t*) curves for TBZ (**a**), amphotericin B (**b**), Daito-*gettou* EO (**c**), and terpinen-4-ol (**d**) obtained using Equation (1).

**Table 1 foods-12-01298-t001:** Minimum inhibitory concentrations (MICs) of antifungal agents and essential oils (Eos) of *gettou* and tea tree against *Aspergillus brasiliensis* NBRC 9455 measured using the broth dilution method.

Antifungal Agent	MIC *		EO	MIC *
	(μg/mL)	(μmol/L)		(%)
Miconazole (MCZ)	10 ^d^	24	Daito-*gettou*	0.40 ^a^
Itraconazole (ITCZ)	10 ^d^	14	Shima-*gettou* 1	0.50 ^a^
Voriconazole (VRCZ)	2.5 ^e^	7.2	Shima-*gettou* 2	0.40 ^a^
Fluconazole (FLCZ)	200 ^b^	650	Shima-*gettou* 3	0.40 ^a^
5-Fluorocytosine (5-FC)	>400 ^a^	>3100	Tea tree	0.20 ^b^
Amphotericin B (AMPH-B)	0.50 ^f^	0.54		
Thiabendazole (TBZ)	50 ^c^	250		

* The MICs were obtained from at least three independent experiments. Data with different letters (a–f) between the antifungal agents or Eos are significantly different (*p* < 0.05).

**Table 2 foods-12-01298-t002:** Chemical compositions of the EOs of *gettou* and tea tree.

No.	Compound *	Relative % Peak Areas **				
		Daito-*gettou*	Shima-*gettou* 1	Shima-*gettou* 2	Shima-*gettou* 3	Tea Tree
1	α-Pinene	1.2	9.6	10.7	1.7	1.4
2	Camphene	0	4.5	13.0	0	0
3	3-Carene	15.1	0	0	10.9	0
4	β-Pinene	2.6	1.6	2.6	2.5	0
5	α-Phellandrene	0	0	1.2	0	0
6	2-Carene	3.6	0	0	8.3	0.9
7	*p*-Cymene	12.3	31.8	28.4	12.4	4.0
8	Limonene	1.6	15.5	31.3	2.4	0
9	β-Phellandrene	0	0	4.9	0	0
10	1,8-Cineole	18.9	11.1	0	22.1	2.2
11	γ-Terpinene	22.2	0.7	0.9	29.5	22.1
12	4-Carene	0	0	0	3.1	2.5
13	Terpinen-4-ol	19.2	0	0	5.6	60.1
14	Caryophyllene	1.6	1.1	1.8	1.4	0
15	Humulene	0	3.8	0	0	0
16	Caryophyllene oxide	1.6	4.2	0	0	0
17	Unknown compound	0	9.4	0	0	0
	Other compounds	0	6.6	5.2	0	6.8
Total		100	100	100	100	100

* Compounds are presented in the order of elution from the Agilent DB-5MS capillary column. ** The gross area of the main peak was calculated as 100%.

**Table 3 foods-12-01298-t003:** Minimum inhibitory concentrations (MICs) of the main components of Daito-*gettou* EO or Shima-*gettou* 1–3 EOs against *Aspergillus brasiliensis* NBRC 9455 measured using the broth dilution method.

Compound *	MIC **	
	(%)	(mmol/L)
α-Pinene	0.60 ^b^	44
Camphene	0.70 ^a^	51
3-Carene	0.70 ^a^	51
*p*-Cymene	0.70 ^a^	52
Limonene	0.60 ^b^	44
1,8-Cineole	0.50 ^c^	32
γ-Terpinene	0.70 ^a^	51
Terpinen-4-ol	0.075 ^d^	4.9

* Compounds used for this measurement were standard products. ** The MICs were obtained from at least three independent experiments. Data with different letters (a–d) between compounds are significantly different (*p* < 0.05).

**Table 4 foods-12-01298-t004:** Chemical components of EOs of *gettou* and tea tree obtained using gas chromatography/mass spectrometry analysis.

Compound	Content * (%)				
	Daito-*gettou*	Shima-*gettou* 1	Shima-*gettou* 2	Shima-*gettou* 3	Tea Tree
*p*-Cymene	8.41	18.47	25.53	9.14	3.53
1,8-Cineole	13.99	9.11	0	16.91	4.85
Terpinen-4-ol	17.24	0	0	8.98	47.54

* The compounds were quantified using the standard product for each compound.

## Data Availability

The data are available from the corresponding author.

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
