# Peer review of "Mode of Antifungal Action of Daito-Gettou (Alpinia zerumbet var. exelsa) Essential Oil against Aspergillus brasiliensis"

_foods, 2023, doi:10.3390/foods12061298_

Round 1
Reviewer 1 Report
Dear authors for manuscript entitled " The Mode of Antifungal Action of the Daito-gettou (Alpinia 2 zerumbet var. exelsa) Essential Oil Against Aspergillus brasiliensis" > I notice a vast effort from the authors. The manuscript is well-organized and structured. It has a good interest to microbiology, drug discovery, and public health readers. The manuscript is generally appropriate and clear in results. Overall, this is a good quality study to illustrate the growth thermograms, which were calorimetric observations for Aspergillus brasiliensis growth in the presence of Daito-gettou, showed a characteristic pattern similar to that of the fungicide amphotericin B, which implies that the mode of action of Daito-gettou may be fungicidal rather than fungistatic.
Here are some comments.
Please write statistical analysis in the M&M
Very important to include figures for the multiplex batch calorimeter with 25 calorimetric units) before and after inoculation with control (fungi alone) and with different treatments. As a new method.
In the step of MIC: Why you chose to dissolve the EO and the antifungal agents in 20% ethyl alcohol not DMSO?
Could you please mention the name of the antifungal agents you used in this step?
Line 132: Could you write the conc. Of ethyl alcohol solution?
Do you have a photos for the inhibition zones that were formed around the EO, antifungal agents ?
Did you measure the inhibition zone around disks inoculated with ethyl alcohol and DMSO each alone (because some times they have antimicrobial effect and they were used as a dissolving agent).
Need to include figures for (the multiplex batch calorimeter with 25 calorimetric units) before and after inoculation with control (fungi alone) and with different treatments.
- Have you described how did you isolate the main components of EO of Daito-gettou 219 and Shima-gettou 1-3?.
- Line 349: could you add this sentence with this reference( Generally, external effect causing change in the ratios between the constituents in the EOs (Nada, H. G., Mohsen, R., Zaki, M. E., & Aly, A. A. (2022). Evaluation of chemical composition, antioxidant, antibiofilm and antibacterial potency of essential oil extracted from gamma irradiated clove (Eugenia caryophyllata) buds. Journal of Food Measurement and Characterization, 1-14)
Where are the statistical results in the result section?
Sincerely,
Author Response
To Reviewer 1
We appreciate the time and effort you have dedicated to providing insightful feedback on ways to improve and enhance our paper.
It is with great pleasure that we resubmit our article for further consideration.
We have incorporated changes that reflect the detailed suggestions you have graciously provided.
We hope that our edits and the responses we provide below satisfactorily address all the issues and concerns you have noted.
To facilitate your review of our revisions, the following is a point-by-point response to the questions and comments delivered.
Thank you for your suggestions. We respond to your comments below.
Please write statistical analysis in the M&M
Response: We added a section on “statistical analysis” to the text. (Lines 214–217)
2.7. Statistical analysis
Statistical analysis was performed using a two-tailed unpaired Student’s t-test (Microsoft Excel 365; Microsoft Corporation, Redmond, WA, USA). A p-value below 0.05 was considered statistically significant.
Very important to include figures for the multiplex batch calorimeter with 25 calorimetric units) before and after inoculation with control (fungi alone) and with different treatments. As a new method.
Response: The figures of the control (fungi alone) were shown at 0 (drug concentration) in Figures 4 and 5. However, as you suggested, we have added the following text to the footnote to make it clearer to the reader. (Lines 591–592)
The concentration “0” in each figure shows the growth thermogram of A. brasiliensis in the broth without antifungal agents.
In the step of MIC: Why you chose to dissolve the EO and the antifungal agents in 20% ethyl alcohol not DMSO?
Response: We chose ethyl alcohol as the solvent of EOs, considering the possibility of applying gettou to foods in the future. DMSO was used because amphotericin B did not dissolve in ethyl alcohol. We added the text as below. (Lines 140–142)
Ethyl alcohol was selected as the solvent in consideration of the future application of Daito-gettou to foods. Amphotericin B, which was insoluble in ethyl alcohol, was dissolved in DMSO.
Could you please mention the name of the antifungal agents you used in this step?
Response: We added the name of the antifungal agents to the text as follows. Please confirm.
(Lines 171–173)
The inhibitory effects of Daito-gettou EO, Shima-gettou 1 EO, terpinen-4-ol, p-cymene, and 1,8-cineole on mycelial growth of A. brasiliensis were measured using the modified "agar dilution method (2007)" for fungi [23].
Line 132: Could you write the conc. Of ethyl alcohol solution?
Response: We felt that the written text was difficult for the reader to understand, so we rewrote it as follows and specifically showed the method. Thank you for pointing this out.
(Lines 173–177)
Each antifungal agent was adjusted to a concentration at 25-, 50-, or 100-fold of the MIC value with ethyl alcohol. The ethyl alcohol solutions (0.15 mL) and the dissolved Sabouraud dextrose agar medium (15 mL) were mixed in a petri dish (φ90 mm) and solidified. The final concentrations of antifungal agents in agar media were adjusted to 1/4, 1/2, and 1/1 MIC values, respectively.
Do you have a photos for the inhibition zones that were formed around the EO, antifungal agents ?
Did you measure the inhibition zone around disks inoculated with ethyl alcohol and DMSO each alone (because some times they have antimicrobial effect and they were used as a dissolving agent).
Response: In this paper, we prepared an agar medium containing EO, placed a filter paper impregnated with funji spores on it, and incubated it. Although the agar medium contains 1% ethyl alcohol, the inhibitory effect of ethyl alcohol is considered negligible because the distance of the grown Aspergillus colony was taken as 100% to calculate the inhibition rate. In addition, we observed that Aspergillus grew normally on the agar medium containing 1% ethyl alcohol.
DMSO was not used because amphotericin B was not used in this measurement.
We are sorry that the description of the method is not sufficient.
Accordingly, we rewrote the section “2.5. Effects on spore germination and mycelial growth”.
We hope our rewrite is clear.
Need to include figures for (the multiplex batch calorimeter with 25 calorimetric units) before and after inoculation with control (fungi alone) and with different treatments.
Response: We would appreciate it if you could confirm below. (Lines 591–592)
The concentration “0” in each figure shows the growth thermogram of A. brasiliensis in the broth without antifungal agents.
Have you described how did you isolate the main components of EO of Daito-gettou 219 and Shima-gettou 1-3?.
Response: We did not isolate the main components of gettou EO but used the standard products.
To clarify this, we have added the following note below Table 3. (Line 428)
Compounds used for this measurement were standard products.
Line 349: could you add this sentence with this reference (Generally, external effect causing change in the ratios between the constituents in the EOs (Nada, H. G., Mohsen, R., Zaki, M. E., & Aly, A. A. (2022). Evaluation of chemical composition, antioxidant, antibiofilm and antibacterial potency of essential oil extracted from gamma irradiated clove (Eugenia caryophyllata) buds. Journal of Food Measurement and Characterization, 1-14)
Response: Thank you for providing a relevant reference for this paper. We have added this reference. (Line 615)
Generally, external effects cause changes in the ratios between the constituents in the EOs [38].
[38] Nada, H. G.; Mohsen, R.; Zaki, M. E.; Aly, A. A. Evaluation of chemical composition, antioxidant, antibiofilm and antibacterial potency of essential oil extracted from gamma irradiated clove (Eugenia caryophyllata) buds. J. Food Meas. Charact. 2022, 16, 673-686. doi: 10.1007/s11694-021-01196-y
Where are the statistical results in the result section?
Response: We have added the statistical analysis results in Figure 2.
We believe that these changes have improved our manuscript. We hope you agree.
Sincerely,

Reviewer 2 Report
The manuscript entitled The Mode of Antifungal Action of the Daito-gettou (Alpinia zerumbet var. exelsa) Essential Oil Against Aspergillus brasiliensis (Okazaki et al.) is very interesting.
This aim of this manuscript was to determine the mechanism of antifungal action of EO of Alpinia zerumbet, commonly known as "ginger peel" in English and "gettou" in Japanese.
The problem of microorganism resistance is also recognized by the WHO. Many of research groups work and want to findinew substances with an antimicrobial effect (including antifungal) in natural resources (sea, plants, etc.) The data obtained are interesting and new.
The authors have adequate conclusions based on the obtained results In my opinion, the manuscript will be improved if the authors include recent literature data.
Line 255: A. brasiliensis should be put in Italic
Author Response
To Reviewer 2
We appreciate the time and effort you have dedicated to providing insightful feedback on ways to improve and enhance our paper.
It is with great pleasure that we resubmit our article for further consideration.
We have incorporated changes that reflect the detailed suggestions you have graciously provided.
We hope that our edits and the responses we provide below satisfactorily address all the issues and concerns you have noted.
To facilitate your review of our revisions, the following is a point-by-point response to the questions and comments delivered.
The authors have adequate conclusions based on the obtained results In my opinion, the manuscript will be improved if the authors include recent literature data.
Response: We have cited recently published papers in the discussion section. [38][48][49]
Line 255: A. brasiliensis should be put in Italic
Response: Thank you for pointing this out.
We have corrected A. brasiliensis in the legend of Figure 2 to italic notation. (Line 494)
We believe that these changes have improved our manuscript. We hope you agree.
Sincerely,

Reviewer 3 Report
Abstract
1. Please add more quantitative data about the main finding especially for the main aroma compounds in the sample by GC MS
2. Please also add more future development of the research
3. More analysis should be fperformed regarding the characteristic of essential oil and using what kind of distillation type ? Is it steam destilation or other approachers ?
Introduction
1. Please add more background about the aims of the research. The determimation of volatile contents was not described
Material and Methods
Please add more explantion about the standard that has been used for getting quantitative data of the component
Results and Discussion
1. Please add statistical notation on the Table and Histogram
Conclusion
Please add brief number about the main research finding
Author Response
To Reviewer 3
We appreciate the time and effort you have dedicated to providing insightful feedback on ways to improve and enhance our paper.
It is with great pleasure that we resubmit our article for further consideration.
We have incorporated changes that reflect the detailed suggestions you have graciously provided.
We hope that our edits and the responses we provide below satisfactorily address all the issues and concerns you have noted.
To facilitate your review of our revisions, the following is a point-by-point response to the questions and comments delivered.
Abstract
- Please add more quantitative data about the main finding especially for the main aroma compounds in the sample by GC MS
Response: Thank you for pointing this out.
We added the following sentences to the abstract. (Lines 23–25)
Terpinen-4-ol content (MIC = 0.075 %) was 17.24 %, suggesting that the antifungal activity of Daito-gettou EO was mainly derived from this component.
- Please also add more future development of the research
Response: Thank you for pointing this out.
We added the following sentences to the abstract. (Lines 29–31)
For confirmation, further studies investigating the molecular mechanisms of this action are needed.
- More analysis should be performed regarding the characteristic of essential oil and using what kind of distillation type ? Is it steam distillation or other approaches ?
Response: We added the following sentences to the abstract and discussion sections. (Lines 18–19)
Therefore, this study aimed to evaluate the antifungal activity of EOs obtained by steam distillation from Daito-gettou leaves …
(Lines 735–738)
In addition, Daito-gettou EO may contain unknown antifungal components, similar to those of Shima-gettou EOs, and therefore, further studies (for example: synergistic effects between EO components, and the influence of EO extraction methods) are needed to probe the mechanisms underlying this activity.
Introduction
- Please add more background about the aims of the research. The determimation of volatile contents was not described
Response: Thank you for pointing this out.
We have included the following sentences to the introduction section. (Lines 54–61)
In addition, their antimicrobial activity depends on the components of the EOs. These spp. contain numerous volatile compounds, such as terpenes, terpenoids, aromatic and aliphatic components, which are naturally synthesized by various parts of the plant as part of their secondary metabolism [7]. The biological properties of EOs are determined by the concentrations of their major components, therefore analyzing their constituents is necessary to assess the biological activity. The components of EOs of Syzygium aromaticumL. Myrtaceae (clove) [16], Citrus tangerina (tangerine), Carum carvi (caraway) [17] and Homalomena pineodora [18] have been analyzed using a gas chromatography/mass spectrometry (GC/MS) technique and reported.
Material and Methods
Please add more explantion about the standard that has been used for getting quantitative data of the component
Response: We added the following sentences to the Materials and Methods section.
(Lines 156–158)
Next, some major components of the EO were quantitatively analyzed. The amount of each component was obtained from a calibration curve prepared using the corresponding standard product.
Results and Discussion
- Please add statistical notation on the Table and Histogram
Response: We added the statistical analysis results in Figure 2.
Conclusion
Please add brief number about the main research finding
Response: We have described as follows. (Lines 735–747)
In this study, the mechanism underlying the antifungal action of Daito-gettou EO was investigated and the following five results were obtained: (1) Daito-gettou EO showed antifungal effects against A. brasiliensis (MIC = 0.4 %). (2) The main chemical components of EO were γ-terpinene, terpinen-4-ol, 1,8-cineole, 3-carene, and p-cymene, which differed from the three kinds of Shima-gettou EOs used in this study. (3) Terpinen-4-ol, which is present in Daito-gettou EO at 17.24 %, showed a higher antifungal activity compared to that of other components (MIC = 0.075 %), and most of the activity of Daito-gettou EO against A. brasiliensis was considered to be contributed by this component. (4) It was revealed that Daito-gettou EO inhibited mycelial growth. (5) The pattern of growth thermograms, which were calorimetric observations of fungal growth in the presence of Daito-gettou EO, was similar to that of the fungicide amphotericin B. These findings imply that the mode of action of Daito-gettou EO is fungicidal, however, to confirm this, further studies investigating the molecular mechanisms of this action are needed.
We believe that these changes have improved our manuscript. We hope you agree.
Sincerely,

Reviewer 4 Report
The manuscript “The Mode of Antifungal Action of the Daito-gettou (Alpinia zerumbet var. exelsa) Essential Oil Against Aspergillus brasiliensis” requires improvement. Authors should revise and improve the English language and grammar throughout the manuscript.
In general, authors should be consistent in using Daito-gettou EOs and gettou EOs throughout the manuscript. Authors should justify on why including tea tree EOs in the present study. Alpha, beta, gamma and etc used in chemical components should be in symbols α, β, γ and etc. A. brasiliensis should be italicised.
Materials and Methods:
Please specify the particular company donated the Daito-gettou EOs. Is it commercially available? The source of the studied EOs is the major concern.
Please cite references on sections 2.3 and 2.4 (https://doi.org/10.1080/0972060X.2018.1526129).
In 2.4, authors mentioned “Quantitative analysis was performed using two method, The mass spectrum of each EO component was determined by comparison with the mass spectrum from the NIST 11 spectrum library. The amount of each component was calculated based on the respective peak areas. In addition, some major components of the EO were quantitatively analyzed and compared with standard products.
o The mass spectrum of each EO component was determined by comparison with the mass spectrum from the NIST 11 spectrum library was not quantitative as mentioned by the authors.
o Authors should be specific in using “amount”. The relative peak area % shown in Table 2 was not quantitatively measured by considering the response factors and the use of standards.
o Major components were mentioned to be quantitatively analysed and shown in Table 4. Please include the quantification method.
o The analysis was performed in how many replicates? There were no SD shown in Table 2.
o No retention indices calculated and compared for each identified components?
Please cite a reference on effects on spore germination and mycelial growth.
On what basis terpinen-4-ol was selected in mycelial growth inhibitory assay? There are other major components as well.
Results:
Please justify on how to group high and low antifungal activity.
This section is rather lengthy. Authors should consolidate this section. Some results were self-explanatory without elaborating much by referring to the Tables and Figures.
Discussion:
This section should be strengthened by comparing the present study with the recent published articles.
Since terpinen-4-ol was assumed to largely contributed to the observed antifungal activity, the discussion around this component should be strengthened. There were many studies conducted on this component.
Chemical structures of major components should be included as well.
Line 405: However, it is assumed that the EO composition of gettou fluctuates with changes in the climate (temperature, rainfall, etc.), which can lead to corresponding fluctuation in the antifungal activity.
o No references was cited nor study was performed to justify this statement.
Author Response
To Reviewer 4
We appreciate the time and effort you have dedicated to providing insightful feedback on ways to improve and enhance our paper.
It is with great pleasure that we resubmit our article for further consideration.
We have incorporated changes that reflect the detailed suggestions you have graciously provided.
We hope that our edits and the responses we provide below satisfactorily address all the issues and concerns you have noted.
To facilitate your review of our revisions, the following is a point-by-point response to the questions and comments delivered.
In general, authors should be consistent in using Daito-gettou EOs and gettou EOs throughout the manuscript. Authors should justify on why including tea tree EOs in the present study. Alpha, beta, gamma and etc used in chemical components should be in symbols α, β, γ and etc. A. brasiliensis should be italicised.
Response: Thank you for your appropriate remarks.
We used Daito-gettou EO and Shima-gettou EO throughout the manuscript.
Also, the reason why tea tree EO was used for comparison in this study is that it is an essential oil with well-known antibacterial activity, and it was thought that its components might be similar to gettou EO. We added the reason in the text as follows. (Lines 90–92)
Tea tree EO was used for comparison in this study because tea tree EO has a high antimicrobial activity [9] and its scent is similar to Daito-gettou EO, thus, it was postulated that the components might be similar.
We have also changed all alpha, beta, and gamma in the manuscript to the symbols α, β, γ. In addition, we corrected A. brasiliensis in the legend of Figure 2 to italic notation.
Materials and Methods:
Please specify the particular company donated the Daito-gettou EOs. Is it commercially available? The source of the studied EOs is the major concern.
Response: We added the name “Kitadaito Island Development Organization.” They manufacture Daito-gettou EO and it is possible to purchase it from them. (Line 111)
Daito-gettou EO used in this study was donated by Kitadaito island development organization, Okinawa Prefecture.
Please cite references on sections 2.3 and 2.4 (https://doi.org/10.1080/0972060X.2018.1526129).
Response: Thank you for providing a relevant reference for this paper. We have added this reference to the introduction section. (Line 59)
The components of EOs Syzygium aromaticum L. Myrtaceae (clove) [16], Citrus tangerina (tangerine), Carum carvi (caraway) [17], and Homalomena pineodora [18] have been analyzed using a gas chromatography/mass spectrometry (GC/MS) technique and reported.
[18] Rozman, N. A. S.; Yenn, T. W.; Tan, W-N.; Ring, L. C.; Yusof, F. A. B. M.; Sulaiman, B. Homalomena pineodora, a novel essential oil bearing plant and its antimicrobial activity against diabetic wound pathogens. J. Essent. Oil Bear. Plants 2018, 21, 963-971. doi:10.1080/0972060X.2018.1526129
In 2.4, authors mentioned “Quantitative analysis was performed using two method, The mass spectrum of each EO component was determined by comparison with the mass spectrum from the NIST 11 spectrum library. The amount of each component was calculated based on the respective peak areas. In addition, some major components of the EO were quantitatively analyzed and compared with standard products.
o The mass spectrum of each EO component was determined by comparison with the mass spectrum from the NIST 11 spectrum library was not quantitative as mentioned by the authors.
o Authors should be specific in using “amount”. The relative peak area % shown in Table 2 was not quantitatively measured by considering the response factors and the use of standards.
o Major components were mentioned to be quantitatively analysed and shown in Table 4. Please include the quantification method.
o The analysis was performed in how many replicates? There were no SD shown in Table 2.
o No retention indices calculated and compared for each identified components?
Response: The relative peak area % shown in Table 2, as you pointed out, was not measured quantitatively.
We modified the text as below.
Quantitative analysis was performed using two methods.
→(Line 153) Analysis was performed using two methods.
The amount of each component was calculated based on the respective peak areas.
→(Line 155) The percentage composition of each component was calculated based on the respective peak areas.
In addition, in the results section “3.1. Antifungal activity”, we have also removed the term meaning quantification.
However, Table 4 shows the quantitatively analyzed results. We modified the quantification method as follows. (Lines 156–158)
Second, some major components of the EO were quantitatively analyzed. The amount of each component was obtained from a calibration curve prepared using the corresponding standard product.
GC/MS measurements were performed multiple times under different conditions, but the measurement under the conditions in this paper was performed once, so there is no SD in Table 2.
Please cite a reference on effects on spore germination and mycelial growth.
Response: We cited the reference "the agar dilution method (2007)" [23]. (Lines 171–173)
The inhibitory effects of Daito-gettou EO, Shima-gettou 1 EO, terpinen-4-ol, p-cymene, and 1,8-cineole on mycelial growth of A. brasiliensis were measured using the modified "agar dilution method (2007)" for fungi [23].
On what basis terpinen-4-ol was selected in mycelial growth inhibitory assay? There are other major components as well.
Response: As shown in Table 3, terpinen-4-ol showed the lowest MIC value among the tested compounds. Therefore, we considered terpinen-4-ol to be the major antifungal component of Daito-gettou EO and selected it for mycelial growth inhibition experiments.
We document our rationale in the discussion section. (Lines 624–659)
Results:
Please justify on how to group high and low antifungal activity.
Response: As you pointed out, it is not possible to distinguish between high and low antifungal activity of EOs. We think that high and low activity would be a matter of comparison. Therefore, we removed all expressions of high or low activity in the manuscript.
We appreciate that you pointed this out.
This section is rather lengthy. Authors should consolidate this section. Some results were self-explanatory without elaborating much by referring to the Tables and Figures.
Response: Thank you for your advice.
We tried to simplify the results by omitting things that can be understood from the table.
We have also merged Sections 3.1 and 3.2 for clarity.
Discussion:
This section should be strengthened by comparing the present study with the recent published articles.
Response: Thank you for your suggestion. We quoted three recently published articles. [38][48][49]
Since terpinen-4-ol was assumed to largely contributed to the observed antifungal activity, the discussion around this component should be strengthened. There were many studies conducted on this component.
Response: We cited and discussed new articles related to the antifungal activity of terpinen-4-ol.
(Lines 713–721)
Chemical structures of major components should be included as well.
Response: We added a figure to the paper showing the chemical structures of the main components. (Figure 1)
Line 405: However, it is assumed that the EO composition of gettou fluctuates with changes in the climate (temperature, rainfall, etc.), which can lead to corresponding fluctuation in the antifungal activity.
o No references was cited nor study was performed to justify this statement.
Response: We apologize for not citing the references.
We cited a reference reporting that the compositions of the EO varied with climate.
(Lines 728–730)
However, it is assumed that the EO composition of gettou fluctuates with changes in the climate (temperature, rainfall, etc.), which can lead to a corresponding fluctuation in its antifungal activity [39].
We believe that these changes have improved our manuscript. We hope you agree.
Sincerely,

Round 2
Reviewer 3 Report
Kindly please provide statistical notation for Table 1, 2 and 3
Author Response
Response to the Comment of Reviewer 3
We thank you for reviewing our manuscript and for providing an insightful comment. We have revised the manuscript accordingly. The following is our response to the comment.
- Kindly please provide statistical notation for Table 1, 2 and 3
Response: We thank you for the comment. We have added the statistical analysis results in Tables 1 and 3.
Regarding the data presented in Table 2, we performed GC/MS measurements several times under different conditions; however, we performed just one measurement under the conditions presented in this manuscript. Therefore, the data in Table 2 cannot be statistically analyzed.

Reviewer 4 Report
Comments are addressed accordingly.
Author Response
Response to the Comment of Reviewer 4
We thank you for reviewing our manuscript and for providing an insightful comment.
The following is our response to the comment.
- English language and style are fine/minor spell check required
Response: We thank you for the comment. We have discussed our results in light of previous studies and added considerations for future studies.
